

# The influence of deep convection on HCHO and $H_2O_2$ in the upper troposphere over Europe

Heiko Bozem[1*], Andrea Pozzer[1], Hartwig Harder[1], Monica Martinez[1], Jonathan Williams[1], Jos Lelieveld[1], and Horst Fischer[1]

[1]Max Planck Institute for Chemistry, POB 3060 55020 Mainz, Germany
[*]now at Johannes-Gutenberg University, Mainz, Germany

*Correspondence to*: Heiko Bozem (bozemh@uni-mainz.de)

**Abstract.** Deep convection is an efficient mechanism for vertical trace gas transport from the Earth's surface to the upper troposphere (UT). The convective redistribution of short-lived trace gases emitted at the surface typically results in a C-
shaped profile. This redistribution mechanism can impact photochemical processes, e.g. ozone and radical production in the UT on a large scale due to the generally longer lifetimes of species like formaldehyde (HCHO) and hydrogen peroxide ($H_2O_2$), which are important $HO_x$ precursors ($HO_x$ = OH+$HO_2$ radicals). Due to the solubility of HCHO and $H_2O_2$ their transport may be suppressed as they are efficiently removed by wet deposition. Here we present a case study of deep convection over Germany in the summer of 2007 within the framework of the HOOVER II project. Airborne in-situ
measurements within the in- and outflow regions of an isolated thunderstorm provide a unique data set to study the influence of deep convection on the transport efficiency of soluble and insoluble trace gases. Comparing the in- and outflow indicates almost undiluted transport of insoluble trace gases from the boundary layer to the UT. The ratios of out/inflow of CO and $CH_4$ are 0.94 ± 0.04 and 0.99 ± 0.01, respectively. For the soluble species HCHO and $H_2O_2$ these ratios are 0.55 ± 0.09 and 0.61 ± 0.08, respectively, indicating partial scavenging and washout. Chemical box model simulations show that post-
convection secondary formation of HCHO and $H_2O_2$ cannot explain their enhancement in the UT. A plausible explanation, in particular for the enhancement of the highly soluble $H_2O_2$, is degassing from cloud droplets during freezing, which reduces the retention coefficient.

## 1 Introduction

Deep convection can transport trace gases from the boundary layer (BL) to the upper troposphere (UT) on time scales of
hours, thus establishing an efficient mechanism for vertical redistribution of trace gases in the troposphere (Gidel, 1983; Chatfield and Crutzen, 1984; Dickerson et al., 1987; Garstang et al., 1988; Pickering et al., 1989; Scala et al., 1990; Lelieveld and Crutzen, 1994; Barth et al., 2015). Especially at mid-latitudes strong zonal winds in the UT accelerate the long range transport of convectively advected trace gases whose sources are in the planetary boundary layer. Thus, reactive gases like nitrogen oxides ($NO_x$ = NO+$NO_2$) and nitrogen compounds that act as $NO_x$ reservoir species, volatile organic
compounds (VOC), oxygenated VOC (OVOC) and carbon monoxide (CO), which have extended lifetimes in the UT, can be



transported over long distances, thus influencing the atmospheric composition and chemistry remote from the sources (Dickerson et al., 1987; Pickering et al., 1996; Jonquieres and Marenco, 1998; Ridley et al, 2004; Bertram et al., 2007, Jaegle, 2007).

The up-lift of ozone precursors by deep convection and the addition of lightning produced $NO_x$ leads to enhanced
photochemical ozone production downwind of thunderstorms (Bozem et al., 2017, and references therein). While insoluble species are efficiently transported to the UT via convection, soluble species are scavenged by cloud and rain droplets, with subsequent removal by precipitation (Wang and Crutzen, 1995; Crutzen and Lawrence, 2000). Nevertheless, recent observations have shown that also highly soluble species can reach the UT via deep convection, most likely due to incomplete removal of these species during precipitation events (Crutzen and Lawrence, 2000; Marie et al., 2000, Barth et
al., 2001; Yin et al., 2002; Borbon et al., 2012; Barth et al., 2016; Bela et al., 2016; Fried et al., 2016). Considerable uncertainty remains with respect to the processes that control the concentrations of soluble species in the outflow of deep convection. Crutzen and Lawrence (2000) emphasize the role of post-convective local photochemical production leading to enhancements of soluble species in the UT, while the study of Barth et al. (2001) identifies dynamical and microphysical processes as the main causes for incomplete removal within clouds. Barth et al. (2001) used a 3D cloud resolving model to
study transport in a mid-latitude storm. Their assumption of incomplete retention of soluble trace gases during freezing of liquid cloud droplets was shown to contribute significantly to transport of soluble species like hydrogen peroxide ($H_2O_2$) to the outflow in the UT, while the assumption of maximum retention both in liquid and ice particles leads to a complete scavenging of $H_2O_2$. Based on a 1D-model study Mari et al. (2000) show that incomplete scavenging of $H_2O_2$ and other species can lead to significant enhancement in the outflow of deep convection. Besides $H_2O_2$, formaldehyde (HCHO)
enhancement was also observed in the outflow of deep convection (Prather and Jacob, 1997; Cohan et al., 1999; Stickler et al., 2006; Fried et al., 2008; Borbon et al., 2012). This may be due to transport from the source region (boundary layer) or to secondary production from convectively transported HCHO precursors like methanol ($CH_3OH$), acetone ($CH_3COCH_3$), acetaldehyde ($CH_3CHO$) and methylhydroperoxide ($CH_3OOH$) (Prather and Jacob, 1997; Fried et al., 2008), which are generally less soluble than HCHO. Stickler et al. (2006) pointed out that increased NO concentrations due to lightning
enhances the HCHO production in the convective outflow. Recently, results from the Deep Convective Clouds and Chemistry (DC3) field campaign (Barth et al., 2015) indicate that, to reproduce the observations, zero ice retention has to be assumed for HCHO and $H_2O_2$, leading to their incomplete removal by precipitation (Barth et al., 2016; Bela et al., 2016; Fried et al., 2016).

Since both $H_2O_2$ and HCHO are precursors of $HO_x$ radicals, transport of these species to the UT in convective clouds has a
significant influence on the oxidizing capacity of the UT, since $HO_x$ production from photolysis of HCHO and $H_2O_2$ typically exceeds primary OH production from $O_3$ photolysis and the subsequent reaction of $O(^1D)$ with water vapour (Jaegle et al., 1997; Prather and Jacob, 1997; Lee et al., 1998; Wang and Prinn, 2000; Mari et al., 2003; Regelin et al., 2013; Lelieveld et al., 2016). Figure 1 illustrates the processes associated with deep convection.



Here we use airborne in-situ measurements taken in the in- and outflow regions of an isolated thunderstorm over south-
eastern Germany on July 19, 2007, to study the influence of deep convection on the transport efficiency of soluble and
insoluble trace gases. Emphasis is given to HCHO, which has a medium solubility described by its Henrys law coefficient of
$k_H = 3.2 \times 10^3$ M atm$^{-1}$, and the highly soluble $H_2O_2$ ($k_H = 8.2 \times 10^4$ M atm$^{-1}$).

## 2 Methods

### 2.1 HOOVER II

The HOOVER project included of a total of two measurement campaigns in October 2006 and July 2007, composed of 4
measurement flights per campaign. From the home airport Hohn (54.2°N, 9.3°E) regular research flights were performed
southbound with a stop-over at Bastia in Corsica (France, 42.2°N, 9.29°E) and northbound with a stop-over at Kiruna airport
(Sweden, 67.5°N, 20.2°E). The majority of the flights were performed in the UT, while regular profiles were flown in and
out of the home and stop-over airports, as well as half-way towards the respective destinations over either southern Germany
or Northern Scandinavia. Additional flights in summer 2007 were directed to the Arctic (Svalbard, Norway, 78.1°N, 15.3°E)
and two flights over central Germany to study the influence of deep convection. On July 19, 2007 an eastward moving
mesoscale convective system developed over the southern part of Germany. During a research flight (HOOVER II flight No.
7) out of Baden airport (48.4°N, 8.4°E) the in- and outflow of a strong convective cell were probed close to Dresden, the
capital of the Free State of Saxony in Germany. Further details about the campaigns can be found in several previous
publications (Klippel et al., 2011; Regelin et al., 2013; Bozem et al., 2017).

### 2.2 Observations

During HOOVER a Learjet 35A from GFD (Hohn, Germany) was used. The jet aircraft has a range of about 4070 km and a
maximum flight altitude of approximately 14 km. In the present configuration, both the range and height maximum were
reduced due to the use of two wing-pods that housed additional instruments. The instrumentation consisted of a
chemiluminescence detector (ECO Physics CLD 790 SR, Switzerland) for NO, $NO_2$ and $O_3$ measurements (Hosaynali Beygi
et al., 2011), a set of up- and downward looking $2\pi$-steradian filter radiometers for J($NO_2$) measurements (Meteorologie
Consult GmbH, Germany), a quantum cascade laser IR-absorption spectrometer for CO, $CH_4$ and HCHO measurements
(Schiller et al., 2008), a dual enzyme fluorescence monitor (model AL2001 CA peroxide monitor, Aerolaser, Germany) to
measure $H_2O_2$ and organic hydroperoxides (Klippel et al., 2011), a laser induced fluorescence (LIF) instrument for
simultaneous measurements of OH and $HO_2$ (Martinez et al., 2010; Regelin et al., 2013), a non-dispersive IR-absorption
instrument (model LI-6262, Li-COR Inc., USA) for $CO_2$ and $H_2O$ measurements (Gurk et al., 2008), a proton transfer
reaction mass spectrometer (PTR-MS, Ionicon, Austria) for partially oxidized volatile organic compounds measurements and




a series of canisters for post flight analysis of non-methane hydrocarbons (Colomb et al., 2006). Details about the instrument performance with respect to time resolution, precision, detection limit and total uncertainty can be found in Klippel et al. (2011), Regelin et al. (2013) and Bozem et al., (2017). For the two species, which are central in the present study, some more details will be given next.

The hydrogen peroxide measurements have a time resolution of 30 sec (time for a calibration signal to rise from 10% to 90%
of total reading), a detection limit of 24 pptv (deduced from the 1-σ reproducibility of in-flight zero air measurements), a precision of ± 8.3 % at 260 pptv (deduced from the 1-σ reproducibility of in-flight calibrations with a liquid standard) resulting in a total uncertainty of ± 13.9 % at 260 pptv (Klippel et al., 2011).

The formaldehyde measurements have a time resolution of 30 sec (averages over ~75 HCHO spectra at a duty cycle of 60 %, while the remainder of the cycle is dedicated to CO and $CH_4$ measurements), a detection limit of 32 pptv (deduced from the
1-σ reproducibility of in-flight zero air measurements) and a total uncertainty of ± 9 % (Klippel et al, 2011; Schiller et al., 2008). All data used in this study have been averaged over a time interval of 30 sec.

## 2.2 Chemistry box model MECCA

In order to estimate photochemical destruction and secondary production of HCHO and $H_2O_2$ in the UT after convective
injection, the box model MECCA (Module Efficiently calculating the Chemistry of the Atmosphere) has been used. MECCA uses an extensive chemistry mechanism for gas phase and liquid phase chemistry (Sander et al., 2005). It uses the "Kinetic PreProcessor" (KPP), a flexible package for the numerical integration to translate the chemical mechanism into a set of ordinary differential equations (Sandu and Sander, 2006). For atmospheric studies, MECCA is coupled to either the atmospheric chemistry-general circulation model EMAC (ECHAM5/MESSy2 Atmospheric Chemistry, www.messy-
interface.org) or the box model CAABA (Chemistry As A Boxmodel Application). The coupling is realised by the MESSy (Modular Earth Submodel System) interface (Jöckel et al., 2006).

For this study CAABA/MECCA was set-up with basic methane, VOC and isoprene chemistry following the Mainz isoprene mechanism (MIM) (Pöschl et al., 2000), a condensed version of the detailed Master Chemical Mechanism (Jenkin et al., 1997). This resulted in a mechanism for tropospheric gas phase chemistry including 76 species and 153 reactions. Halogen
chemistry and liquid phase or heterogeneous chemistry was not included, since model calculations were restricted to cloud free air. Additionally, deposition processes, either wet or dry, were not considered for the same reasons. Equations were solved by a Rosenbrock scheme (Sandu and Sander, 2006). Details of the model set-up and initialization will be discussed in section 3.3.



## 3 Results

### 3.1 Meteorology on July 19, 2007


The meteorological conditions in Europe on July 19, 2007 are shown in Fig. 2. A pool of cold air, associated with a long trough extending from the North Atlantic over the British Isles to the Azores, led to a low pressure system between Spain and Ireland (Fig. 2a). Towards the east, a high pressure ridge extending from North Africa over the eastern Mediterranean,

Eastern Europe and into Russia, started to weaken. Between these systems a strong south-westerly flow was established in the middle and upper troposphere, bringing moist and warm air from Spain to north-western Europe, with a surface frontal zone, separating moist, hot air in the south-east from dry, cold air in the north-west (Fig.2b). Over Germany these conditions lead to the development of thunderstorms, favoured by a potentially unstable troposphere with high CAPE (Convective Available Potential Energy) values over south-western Germany.

During the night from 18 to 19 July convective cells developed along a weakening cold front over France, which rapidly developed into a mesoscale convective system (MCS) moving into south-western Germany in the early morning hours of July 19. This MCS subsequently travelled in north-easterly direction accompanied by heavy rain (Fig. 3). Around noon the south-eastern edge of the MCS reached the Nürnberg/Bamberg area. The explosive storm development came with three strong, isolated cells which expanded throughout the whole tropospheric column up to 10 km altitude within two hours. The

convection in the three cells was accompanied by strong lightning activity. Figure 4 shows the position (Fig. 4a) and intensity (Fig. 4b) of detected lightning flashes on July 19, 2007 between 0 UTC and 22 UTC. Based on the temporal evolution of the lightning activity (colour code in Fig. 4a) the movement of the MCS over Europe can be traced up to the rapid thunderstorm development over south-eastern Germany (green and yellow points in Fig. 4a). During this phase the number of lightning flashes increased to 50 flashes/min (Fig. 4b).

Figure 5 (a-c) shows the RADOLAN (RADar-OnLine-ANeichung) data of the precipitation radar of the German weather service (DWD) for the time period of rapid development over the Nürnberg/Bamberg area. During the initial storm development (Fig. 5a), from 11:30 UTC to 12:30 UTC, precipitation is observed associated with the MCS. Later on (12:30 UTC to 13:30 UTC; Fig. 5b) the isolated thunderstorm cells, being investigated here, emerged. Superimposed on the figure is the flight track of the Learjet after take-off from Baaden airport to the UT ahead of the MCS. The precipitation signal from

three not fully separated thunderstorm cells in the time interval 13:30 UTC – 14:30 UTC together with the flight track of the Learjet is shown in Fig. 5c. The strongest cell at the most northerly position with a maximum precipitation intensity of 30 mm/h was intensively probed by the research aircraft, performing measurements around the cell at various altitudes in, above and below the outflow during the final period (14:30 UTC to 15:30 UTC) (Fig. 5d). The outflow, which occurred at an altitude of 10.5 km, was identified in-flight by the strong enhancements of a number of trace gases (see section 3.2) and

sampled for about 5 min. Due to the close proximity of the three cells the outflow cannot be assigned unambiguously to an individual cell, but on-board wind measurements strongly support our interpretation that the outflow originated from the most northern cell.



The research flight was continued by measurements ahead of the weakening MCS. Around Dresden descents into the continental boundary layer were flown, followed by ascents, to obtain trace gas profiles and to characterize the potential

inflow region, spending 15 min in the boundary layer. Although the inflow region cannot be identified unambiguously, the storm track and local winds support our interpretation.

### 3.2 Observations

Figure 6 shows a time series of trace gas measurements ($CH_3OH$, $CH_3COCH_3$, $HO_2$, OH, NO, $O_3$, $H_2O_2$, HCHO, $CH_4$, CO

and flight altitude) from HOOVER flight 7 on July 19, 2007. The first part of the flight was conducted in the wake of the eastward moving MCS and up-wind of the developing thunderstorm cells. Signatures of convection can be identified during various parts of the flight. First significant enhancements in CO and $CH_4$ between 13:00 UTC and 13:10 UTC at an altitude between 7.5 and 8 km were influenced by activity of the MCS that moved over the area earlier during the day. Signatures of recent convection can also be identified from enhancements of $O_3$ and $CH_3COCH_3$, while mixing ratios of shorter lived

species e.g. NO, HCHO, $H_2O_2$ and $HO_x$-radicals are only slightly influenced. The lack of enhancements of these species and the enhancement of $O_3$ point towards ozone build-up, and indicate that we most likely probed an aged air mass from an up-wind convective event that had occurred recently (DeCaria et al., 2005).

Signatures of rather fresh convective injection into the UT are found between 13:45 UTC and 13.58 UTC, with enhancements above background concentrations, observed for $CH_3OH$, $CH_3COCH_3$, NO, $H_2O_2$, $CH_4$ and CO and a slight

decline of $O_3$ mixing ratios, indicating recent transport from lower layers of the troposphere. Formaldehyde does not show clearly enhanced mixing ratios. The strong increase of NO indicates production by lightning, which was directly observed in the area (see Fig. 4).

The major convective event associated with the northernmost thunderstorm cell was probed between 14:27 UTC and 14:42 UTC (marked by the red box in Figure 6). Before entering the outflow, but in close proximity to the convective cell, all trace

gases with the exception of $O_3$ showed significant decreases in mixing ratios. Between 14:10 UTC and 14:27 UTC CO and $CH_4$ dropped to mixing ratios around 65 ppbv and 1770 ppbv, respectively, while $O_3$ increased to 130 ppbv, indicating downward transport from the stratosphere. PV-maps indicate that the local tropopause was about 1-2 km above the aircraft flight track. It has been postulated by Poulida et al. (1996) and later demonstrated by Pan et al. (2014) that convective cells in the mid-latitudes are often surrounded by $O_3$ rich air masses associated with transport from the stratosphere, and this

phenomenon was also observed during this flight.

As mentioned above, the outflow of the northernmost cell was probed between 14:27 UTC and 14.42 UTC. Here all trace gases and radicals under investigation show significant changes in mixing ratios. Table 1 lists the changes in mixing ratios in the outflow, the surrounding UT and the enhancement ratio (outflow/surrounding air). The mixing ratios for the surrounding air masses were determined from a part of the flight around 14:15 UTC up-wind of the thunderstorm and before the period



190 affected by the stratospheric influence described above. Positive values for enhancement ratios are reported for all species except $O_3$ with a ratio less than 1. These enhancements vary between 4 % for $CH_4$ and 700 % for NO, while $O_3$ in the outflow was 20 % lower than in the surrounding air masses. Even higher enhancement ratios are observed for HCHO, though also related to the uncertainty in the mixing ratio for the surrounding air mass that was below the detection limit of the instrument. Besides HCHO, the strongest enhancement is observed for NO, most probably due to lightning.

195 After probing the outflow, the aircraft performed a descent into the boundary layer ahead of the thunderstorm over Dresden (marked by the blue box in Fig. 6). Vertical profiles from this part of the flight (between 14:20 UTC and 15:25 UTC) are shown in Fig. 7. Visual inspection of Fig. 7 indicates that mixing ratios for the longer lived, insoluble trace gases CO, $CH_4$, $CH_3COCH_3$ and $CH_3OH$ in the outflow are of the same order of magnitude as in the inflow area, i.e. the boundary layer towards the north-east. The mixing ratios of NO, $O_3$ and OH in the outflow are higher than in the inflow, while for $HO_2$,

200 HCHO and $H_2O_2$ we find the reverse. The influence of the high NO on $HO_x$ partitioning has been addressed by Regelin et al. (2013) and will not be discussed here. From the behaviour of the longer-lived, insoluble tracers and assuming that the measurements in the boundary layer are representative for the in-flow of the thunderstorm cell, we infer that the convection transported nearly undiluted boundary layer air into the UT. Table 2 shows that the median (mean ± 1-σ standard deviation) ratios of outflow to inflow mixing ratios for CO and $CH_4$ were 0.93 (0.94 ± 0.04) and 0.99 (0.99 ± 0.01), respectively, and

205 thus close to unity. For simplicity we therefore assume that the contribution of entrainment is insignificant for the outflow region. Thus the lower than unity values for outflow/inflow ratios for HCHO and $H_2O_2$ of 0.54 (0.55 ± 0.09) and 0.59 (0.61 ± 0.08), respectively, are most likely due to partial rain-out of these soluble species. As illustrated in the schematic in Fig. 8 the 5 min measurements in the outflow were made in clear, cloud free air at a distance between 50 and 150 km from the anvil. At a wind speed of 30 m/s this would correspond to a transport time of 30 to 90 min which is sufficient for secondary

210 photochemistry (production or destruction) to become a significant contributor to the budgets of shorter lived species like HCHO and $H_2O_2$. For example, Bozem et al. (2017) calculated the potential net ozone production rate from in-situ observation in this convective event and derived a rate of 1.9 ± 0.28 ppbv/h, indicating secondary ozone formation of 1 − 3 ppbv in the outflow, explaining the slight difference in $O_3$ between the inflow (80.8 ppbv; 81.2 ± 2.7 ppbv) and outflow region (83.8 ppbv; 83.5 ± 2.6 ppbv) (Table 2).

215 As shown by Fried et al. (2008), the temporal evolution of the HCHO mixing ratio depends on the concentration of HCHO at the cloud top, the concentrations of HCHO precursors and radicals, and the processing time. Within the first few hours it is unlikely that the HCHO concentration will reach steady state. Here we will use a box model to simulate the temporal evolution of HCHO and $H_2O_2$ mixing ratios in the outflow region after exiting the cloud. Please note that this model study neglects mixing with background air. This is justified by the ratios between inflow and outflow given in table 2 for the

220 longer lived species, which are close to unity, indicating insignificant mixing with surrounding air masses.




### 3.3 Temporal evolution of HCHO and H$_2$O$_2$ mixing ratios

To calculate the temporal evolution of HCHO and H$_2$O$_2$ with the MECCA model, we constrained it with measured median OH, HO$_2$, CO, CH$_4$, CH$_3$OH, CH$_3$COCH$_3$, O$_3$ and NO mixing ratios, and photolysis rates derived from the radiation transfer model TUV (https://www2.acom.ucar.edu/modeling/tropospheric-ultraviolet-and-visible-tuv-radiation-model (Madronich and Flocke, 1999)), based on observed J(NO$_2$) frequencies. Calculations of chemical production and loss were performed for two hours with a time step of 15 min. In general the model runs indicate a strong dependency of the mixing ratio evolution on processing time and the amount of HCHO and H$_2$O$_2$ leaving the convective cloud (initial values). At high HCHO and H$_2$O$_2$ initial concentrations photochemical destruction due to photolysis and reaction with OH prevails (the loss terms are proportional to the initial concentration), while at low initial concentrations secondary production from precursors dominate. Therefore, a number of sensitivity studies were performed that take into account the uncertainties in initial values and processing time. To account for variations in the input parameters and uncertainties of the rate constants of the model we performed additional sensitivity calculations based on the Monte Carlo method, by varying initial concentrations within the 1σ uncertainties and rate constants by up to ± 80 %.

Figure 9 shows the temporal evolution of HCHO mixing ratios in the outflow as simulated with MECCA. The blue area (median with standard deviation) indicates the observed HCHO concentration in the outflow (1.45 ± 0.11 ppbv). Sensitivity studies were performed with different initial HCHO concentrations. A first run was performed with an initial HCHO mixing ratio of 0.02 ppbv, corresponding to background conditions (Table 1). This case study represents near-complete removal of HCHO during convective uplifting due to cloud processing, followed by subsequent secondary production from HCHO precursors in the cloud free outflow. Note that all precursors and radical levels are initialized at observed values in the outflow and do not change during the processing. The red curve in Figure 9 starts at background HCHO mixing ratios, and exhibits significant production of HCHO during the first 60 min with a rate of approx. 0.01 ppbv/min with a maximum HCHO mixing ratio of 0.56 ppbv (range 0.51 – 0.61 ppbv), corresponding to the minimum and the maximum of the Monte Carlo simulation) after around 90 min, and slowly decreasing values afterwards. These results compare well to box model studies of secondary HCHO formation after convective events reported by Stickler et al. (2006) and Fried et al. (2008). Since the mixing ratios in this simulation are at any time smaller than values observed in the outflow, we can assume that a significant portion of HCHO observed in the outflow is due to vertical transport from the inflow region.

A second run (not shown) was performed with an initial HCHO mixing ratio corresponding to the observed value in the outflow (1.45 ± 0.11 ppbv). Photochemical loss of HCHO through photolysis and reaction with OH dominates the temporal evolution of formaldehyde in this run, with loss rates up to 0.02 ppbv/min. This case study demonstrates that HCHO initial values have to be larger than observed mixing ratios in the outflow if we take photochemical processing into account. On the other hand, if we fully neglect photochemical processing by setting the processing time to zero, the observed concentration in the outflow is identical to the amount transported upwards in the convective cloud, providing a lower limit for the cloud top outflow. In a final sensitivity study we assumed undiluted transport of HCHO from the boundary layer to the cloud top.



Starting with an initial value of 2.70 ± 0.42 ppbv corresponding to the HCHO mixing ratio in the inflow the green curve in

255   Figure 9 shows strong photochemical loss of HCHO throughout the simulation. The envelope given by the minimum and maximum of the initial concentrations and the Monte Caro simulation intercepts the observed HCHO mixing ratio range between 20 and 52 min (vertical dashed lines). Due to both the uncertainty of the HCHO initial values and the elapsed processing time it is not possible to assign a single value to the HCHO mixing ratio in the cloud outflow. Instead we used a range corresponding to a minimum value given by the measured HCHO mixing ratio in the cloud free outflow (1.45 ± 0.11

260   ppbv) and assuming zero processing and a maximum value assuming undiluted HCHO transport from the inflow region (2.70 ± 0.42 ppbv) and a processing time of 20 to 52 min.

Figure 10 shows the temporal evolution of $H_2O_2$ in the outflow region calculated with MECCA. Independent of the initial $H_2O_2$ mixing ratio all simulations indicate a photochemical loss of hydrogen peroxide in the first 2 hours after convective injection. Assuming a processing time between 20 and 52 min deduced from the HCHO study (vertical dashed lines), the

265   best fit for the initial $H_2O_2$ concentration is obtained for a range between 1.42 and 1.45 ppbv (brown trace in Figure 10). Lower initial values (red trace for background conditions corresponding to zero transport) or higher values (green trace for complete transport from the inflow region to the outflow) are at no time compatible with the observations (blue bar). As in the case of HCHO we will again provide a range of $H_2O_2$ initial values that are compatible with the observations in the outflow, with a minimum of 1.25 ± 0.09 ppbv given by the observed mixing ratio and no photochemical processing and a

270   maximum of 1.435 ± 0.015 ppbv provided for the model best fit at processing times between 20 and 52 min. It should be mentioned that the MECCA simulations are based on gas phase chemistry only. Hydrogen peroxide formation in the liquid phase by $HO_2$ or $O_3$ dissolution is ignored. Based on short transport times from the boundary layer to the UT of the order of 30 min at vertical velocities of 5 m/s and a depth of 9 km, the $H_2O_2$ production by these processes is negligible (Prather and Jacob, 1997; Jacob, 2000).

275

## 3.4 Estimation of scavenging efficiencies for HCHO and $H_2O_2$

Based on the model results presented above, we estimate the HCHO mixing ratio at the cloud top exit to be in the range between 1.45 and 2.70 ppbv. This corresponds to the amount of HCHO transported from the inflow region to the top of the cloud. Assuming no chemical processing in the cloud, the ratio between the modelled cloud top mixing ratio and the mixing

280   ratio in the inflow region yields the transport efficiency for HCHO at a range of 53 – 100 %. This indicates a strong contribution of transport to the HCHO budget in the outflow, which corresponds to minor cloud scavenging, i.e. at an efficiency between 0 and 47 % (amount of HCHO lost within the cloud). For $H_2O_2$ a similar analysis yields a cloud top mixing ratio of 1.25 to 1.45 ppbv, a transport efficiency of 59 to 68 % and a corresponding scavenging efficiency of 32 to 41 %.

285



## 4 Discussion and Conclusions

The main result of this study is that HCHO and $H_2O_2$ observed in the outflow of a deep convective cloud in the UT are largely controlled by transport from the lower troposphere. Post-convective photochemical processing in cloud free air cannot explain the observations of both species since chemical loss processes are found to dominate. This means that at least 53 % of HCHO and at least 59 % of $H_2O_2$ from the boundary layer reach the cloud top. These percentages increase further if we take into account photochemical processing during a time period of 20 to 52 min, yielding 100 % (HCHO) and 68 % ($H_2O_2$), respectively. This is a consequence of the dominance of loss processes after injection into the outflow region. Formaldehyde is more sensitive to the post cloud processing due to its shorter photochemical lifetime, resulting in a larger range of potential initial values that are compatible with the observations. From these results we deduced scavenging efficiencies for HCHO and $H_2O_2$ of 0 to 47% and 32 – 41%, respectively.

Several studies reported in the literature show that HCHO in UT convective outflow can be significantly enhanced. Previous observation-based studies have attributed this HCHO enhancement either completely (Stickler et al., 2006; Fried et al., 2008a) or largely (60 %, Borbon et al., 2012) to secondary photochemical production in the outflow. The studies of Stickler et al. (2006) and Fried et al. (2008a) involved substantial distance from the convective outflow, and represented extensive processing of air within the UT. This might explain the high contributions for secondary production found in these studies, while we simulate strong photochemical loss of HCHO in the first two hours after convective injection. Studies by Mari et al. (2000), Barth et al. (2001) and Mari et al. (2003) were based on model simulations and observations, and addressed the influence of convective transport on the budget of HCHO in the UT. They emphasised the importance of HCHO scavenging and discussed the effect of incomplete retention by hydrometeors during freezing. Barth et al. (2007) considered gas, liquid and frozen water chemistry, and estimated scavenging efficiencies for HCHO of 46 – 67 %. This is higher than the range presented here, which agrees better with the findings of Borbon et al. (2012), who derived a very small scavenging efficiency of 4 ± 1 % for a MSC storm over a tropical forest region of Oueme and values of 26 ± 8 %, 39 ±12 % and 13 ± 4 % for three storms over other regions of West Africa. Analysing data from a number of storms over North America as part of the DC3 aircraft campaign, Fried et al. (2016) derived scavenging efficiencies of 54 ± 3 %, 54 ± 6 %, 58 ± 13 % and 41 ± 4 % for four storms in May and June 2012, which is again higher than our result (range 0 – 47 %).

The scavenging efficiency for $H_2O_2$ of 32 – 41 % deduced in this study is lower than most values reported in the literature thus far. Based on 3D-model results Barth et al. (2007) report a range between 55 % and 65 %. From in-situ observations in DC3, $H_2O_2$ scavenging efficiencies between 79 and 97 % were deduced (Barth et al., 2016, Bela et al., 2016). As has been shown by a number of model studies (Mari et al., 2000; Barth et al., 2001; Mari et al., 2003, Bela et al., 2016) the $H_2O_2$ scavenging efficiency strongly depends on the fate of $H_2O_2$ during freezing of cloud particles. Incomplete retention can lead to degassing of $H_2O_2$ from the droplets and reduce scavenging efficiencies. The retention coefficient describing the fraction of a dissolved species retained in the droplet during freezing is highly uncertain. Reported values in the literature for $H_2O_2$ retention vary between 5 % and 100 % (Iribarne and Pyshnov, 1990; Snider et al., 1992; Conklin et al., 1993; Snider and





Huang, 1998). Experiments in the Mainz vertical wind tunnel lab yielded a $H_2O_2$ retention coefficient of $52 \pm 8$ % (von
320   Blohn et al., 2011), which is about 10 to 20 % higher than our results. It should be mentioned that such an analysis will also
depend on the ice fraction in the clouds that are typically mixed-phase systems. Ice fractions are expected to vary so that
retention coefficients may fluctuate accordingly.

Overall our results compare well to literature values, with scavenging efficiencies for both HCHO and $H_2O_2$ being at the
lower end of those reported. The model calculations in section 3.3 indicate that the temporal evolution of both HCHO and
325   $H_2O_2$ after convective injection depends strongly on the initial values and the processing time. In particular during the first
120 min, when both species are far off from photostationary state, changes are very large and give rise to large uncertainties.
Fried et al. (2016) also pointed out that the definition of inflow and outflow regimes can be critical. They report a case of
weak convection, and their analysis provided a rather high scavenging efficiency for HCHO of $81 \pm 5$ %, attributed to a
mismatch between in- and outflow of the system. In our study it is not possible to unambiguously identify the inflow area.
As shown by Klippel et al. (2011) the boundary layer mixing ratios for HCHO and $H_2O_2$ near Dresden are similar to other
boundary layer observations during HOOVER II, so that we consider them as representative for the area. The measured
vertical profiles indicate strong variations (Fig. 7 and (Klippel et al., 2011)), with higher $H_2O_2$ and lower HCHO mixing
ratios above the boundary layer. Inflow from above the boundary layer would lead to lower/higher scavenging efficiencies of
HCHO and $H_2O_2$. Nevertheless, analysis of the conserved tracers such as CO and $CH_4$ indicate almost undiluted transport
from the boundary layer to the UT, thus with a negligible contribution of entrainment.

*Acknowledgements.* The authors would like to acknowledge the support from the HOOVER team, envisvcope GmbH
(Frankfurt), and GFD (Gesellschaft für Zielflugdarstellung, Hohn).






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



**Table 1: Median mixing ratios of different species in the outflow region of the thunderstorm cloud and its surroundings, including the enhancement ratios.**

| Species | Mixing ratio outflow region (ppbv) | Mixing ratio surrounding (ppbv) | Ratio (outflow region)/(surrounding) |
|---|---|---|---|
| CO | 118.5 | 63.7 | 1.86 |
| CH$_4$ | 1852.8 | 1785.6 | 1.04 |
| HCHO | 1.45 | 0.02 | 90 |
| O$_3$ | 83.8 | 104.8 | 0.80 |
| H$_2$O$_2$ | 1.25 | 0.91 | 1.37 |
| HO$_2$ (pptv) | 6.63 | 2.71 | 2.44 |
| OH (pptv) | 3.3 | 0.73 | 4.51 |
| NO | 0.96 | 0.12 | 7.75 |
| Acetone | 2.84 | 0.64 | 4.44 |


**Table 2: Mixing ratios observed in the out- and inflow regions, and their ratios. The median, mean and 1σ standard deviation of all measurements is listed.**

| Species | Mixing ratio outflow region (ppbv) | Mixing ratio inflow region (ppbv) | Ratio (outflow region)/(inflow region) |
|---|---|---|---|
| CO | 118.5 (119.8 ± 3.9) | 127.8 (127.5 ± 3.0) | 0.93 (0.94 ± 0.04) |
| CH$_4$ | 1852.8 (1853.1 ± 12.0) | 1876.7 (1876.4 ± 10.4) | 0.99 (0.99 ± 0.01) |
| HCHO | 1.45 (1.47 ± 0.11) | 2.70 (2.69 ± 0.42) | 0.54 (0.55 ± 0.09) |
| H$_2$O$_2$ | 1.25 (1.28 ± 0.09) | 2.11 (2.09 ± 0.21) | 0.59 (0.61 ± 0.08) |
| O$_3$ | 83.8 (83.5 ± 2.6) | 80.8 (81.2 ± 2.7) | 1.04 (1.03 ± 0.05) |
| NO | 0.96 (0.99 ± 0.20) | 0.05 (0.05 ± 0.03) | 19.2 (19.8 ± 12.54) |
| OH (pptv) | 3.3 (2.91 ± 0.93) | 0.28 (0.26 ± 0.13) | 11.79 (11.19 ± 6.64) |
| HO$_2$ (pptv) | 6.63 (6.04 ± 1.04) | 16.94 (18.41 ± 3.08) | 0.39 (0.33 ± 0.08) |
| Acetone | 2.84 (2.82 ± 0.21) | 2.33 (2.30 ± 0.22) | 1.22 (1.23 ± 0.12) |
| Methanol | 7.19 (7.12 ± 0.36) | 7.71 (7.63 ± 0.64) | 0.93 (0.93 ± 0.10) |






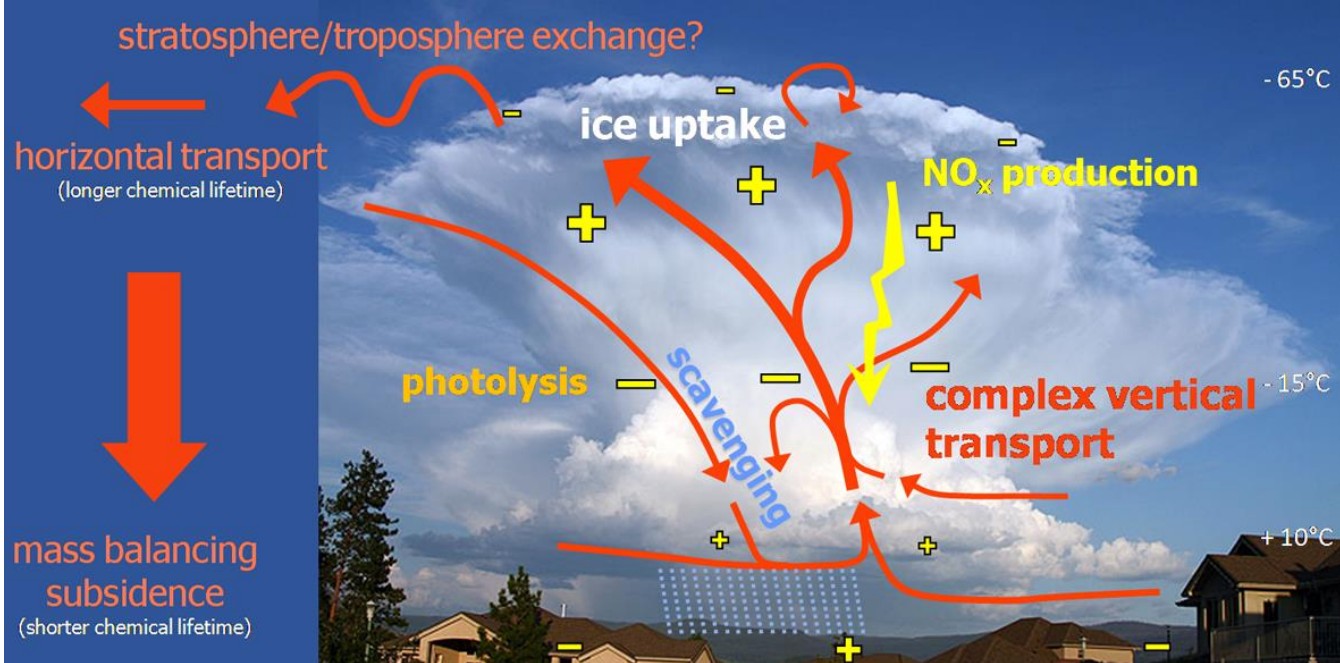

**Figure 1: Scheme of the relevant characteristics and processes of a thunderstorm cloud.**


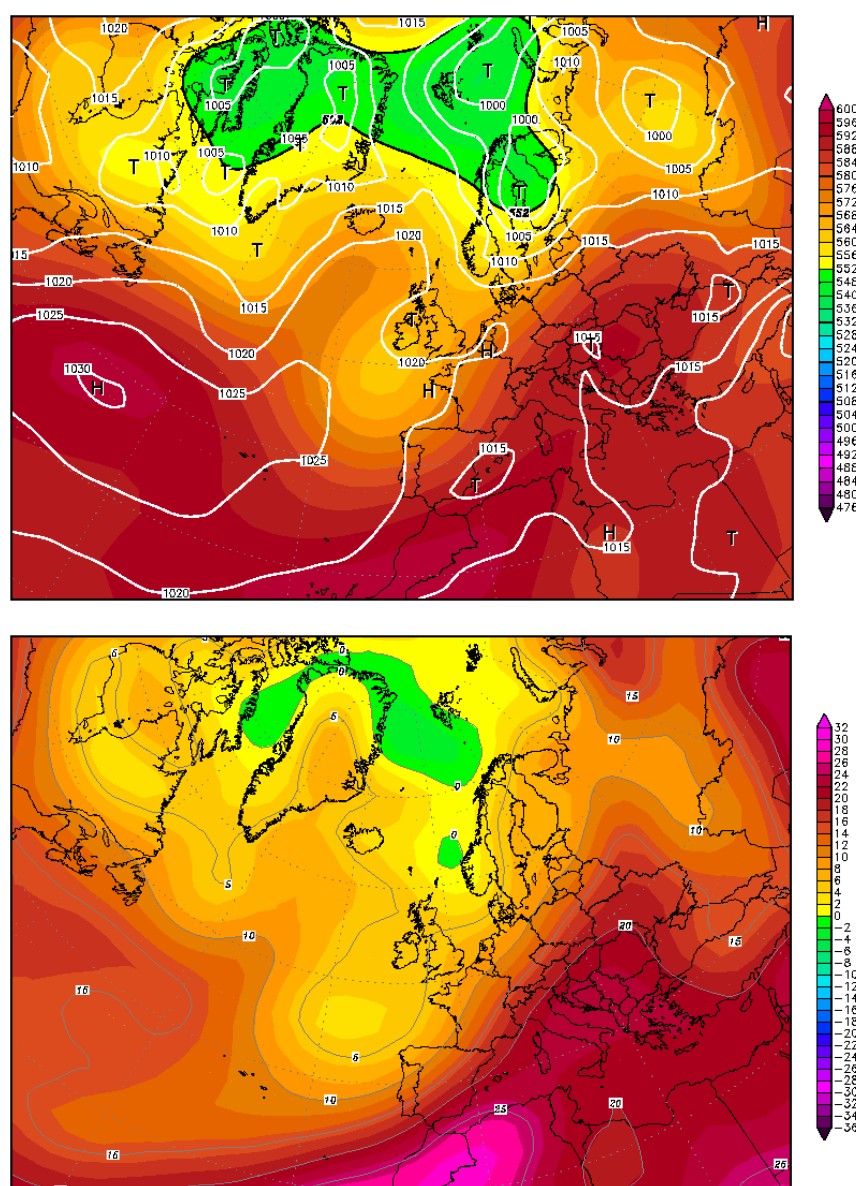


**Figure 2: NCEP reanalysis data for 19th July 2007. (top): 500hPa geopotential height (colour scale in m). The white lines mark the surface pressure field (in hPa). (bottom): Temperature distribution at 850 hPa (°C).**





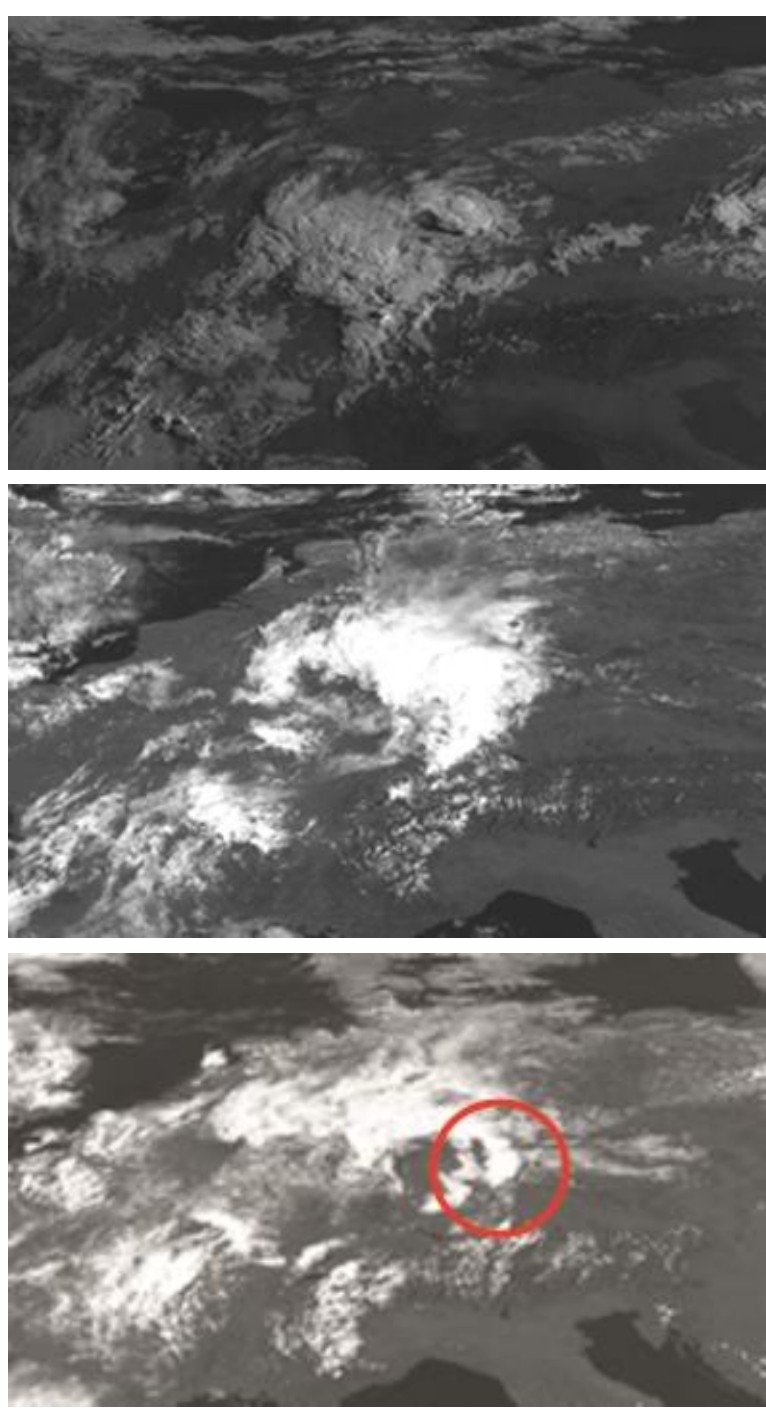


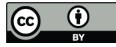



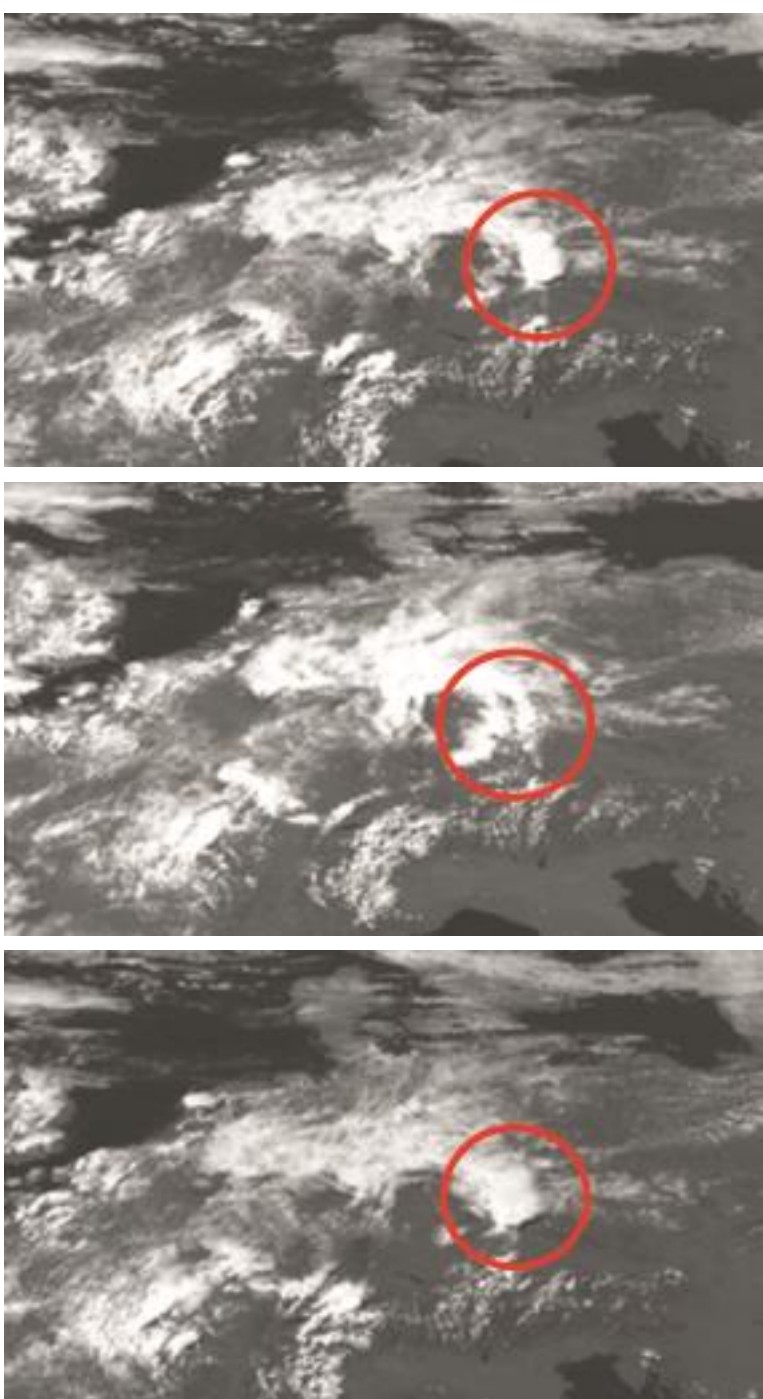


**Figure 3: Temporal evolution (6:00 UTC, 10:00 UTC, 13:00 UTC, 13:30 UTC, 14:00 UTC, and 14:30 UTC from top to bottom) of the MCS over Germany observed from Meteosat, obtained from the High-Resolution Visible (HRV) channel.**



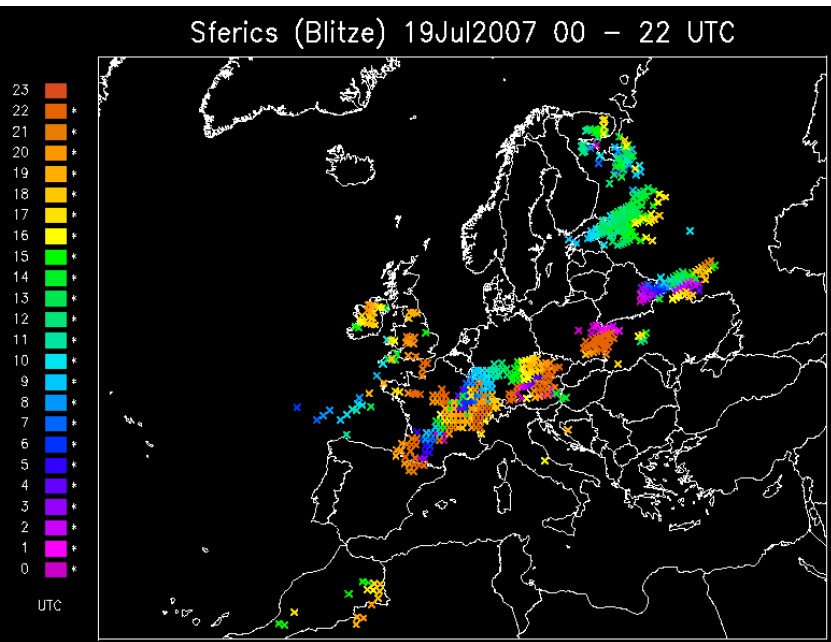

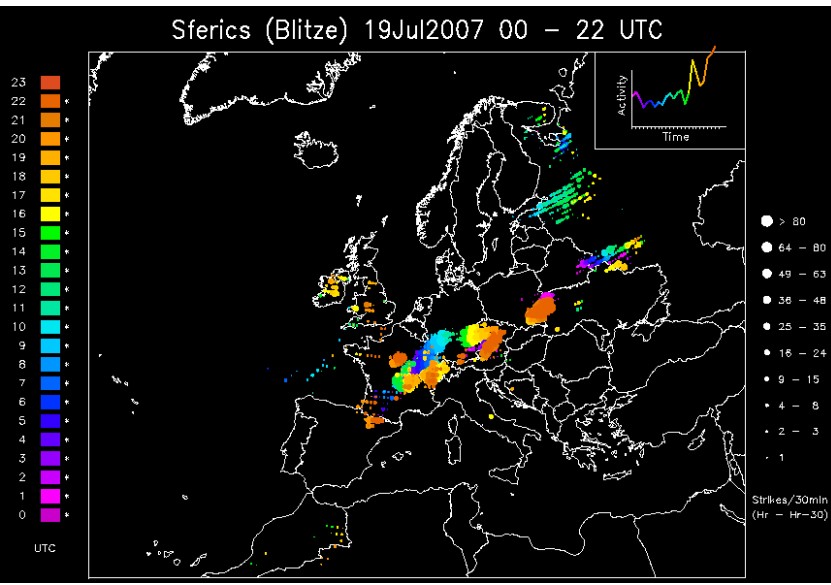


**Figure 4: (top) Lightning activity from 0UTC to 22UTC on 19th July 2007 over Europe. (bottom) Lightning flash occurrence in lashes per 30min for the same time interval. The colour code indicates the time of flash detection. Source: www.wetterzentrale.de.**






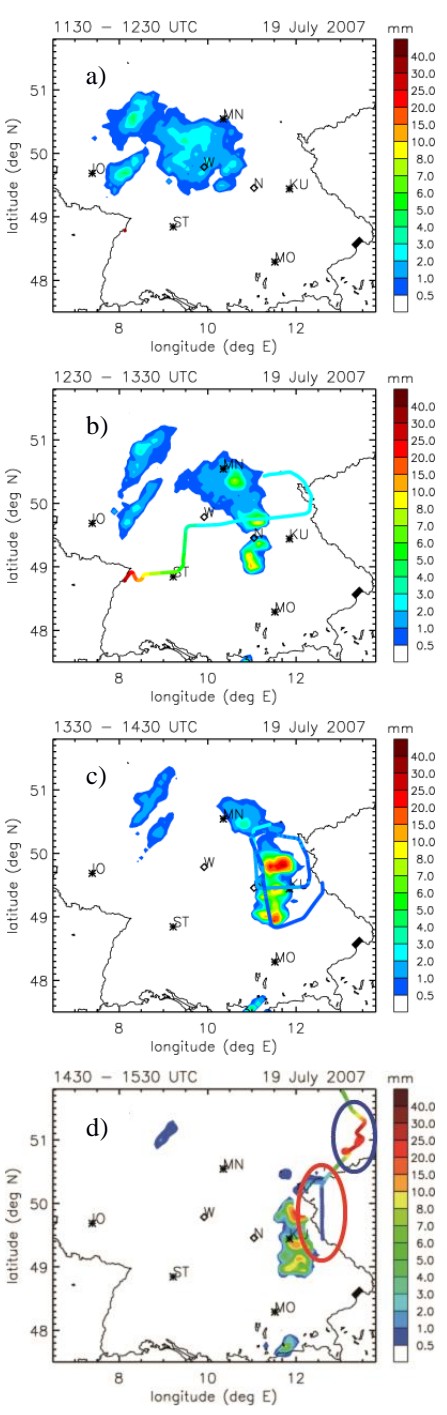

**Figure 5: (a-d): RADOLAN data of the German weather service. Precipitation intensity is colour coded in mm per hour. The flight tracks are also shown (red: low flight levels, blue: high flight levels). The red marked area in the lower panel shows the outflow region, the blue one the inflow region in the boundary layer nearby Dresden. Data were provided by M. Zimmer with permission of E. Weigel (German weather service).**





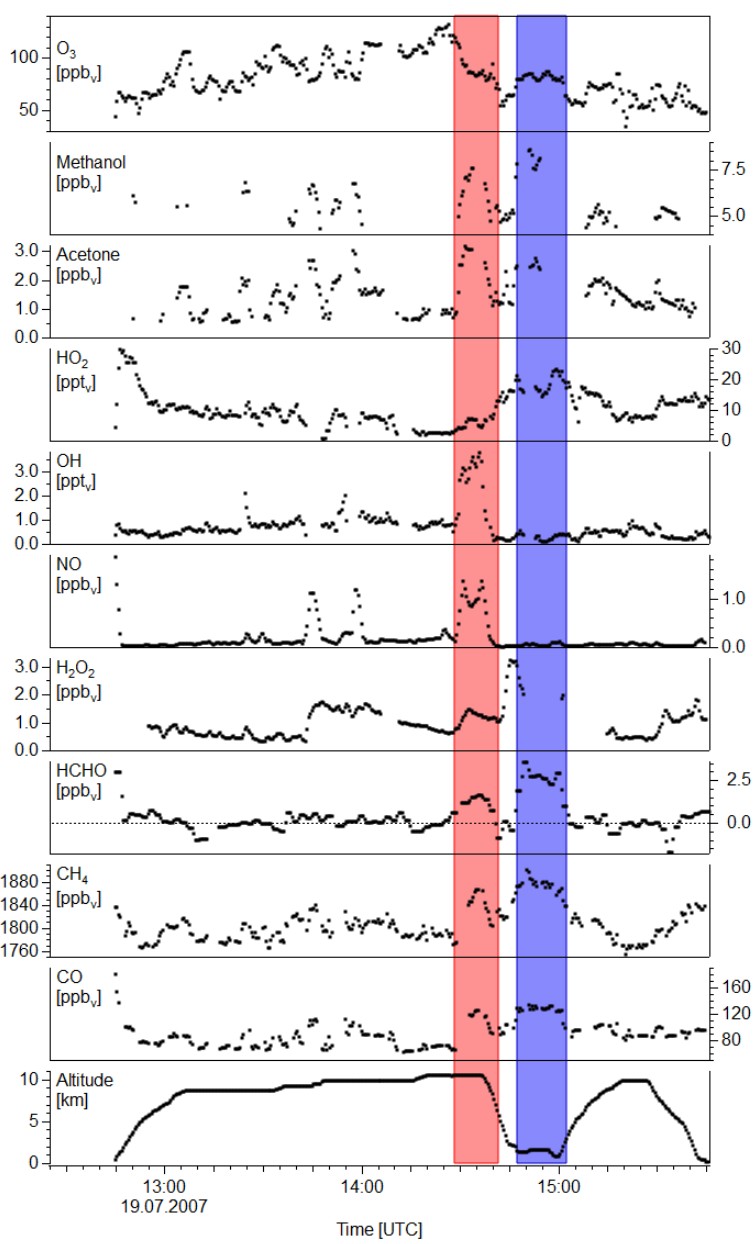


**Figure 6: Time series of parameters measured on 19th July 2007. The red area marks the outflow region, the blue area the inflow region.**





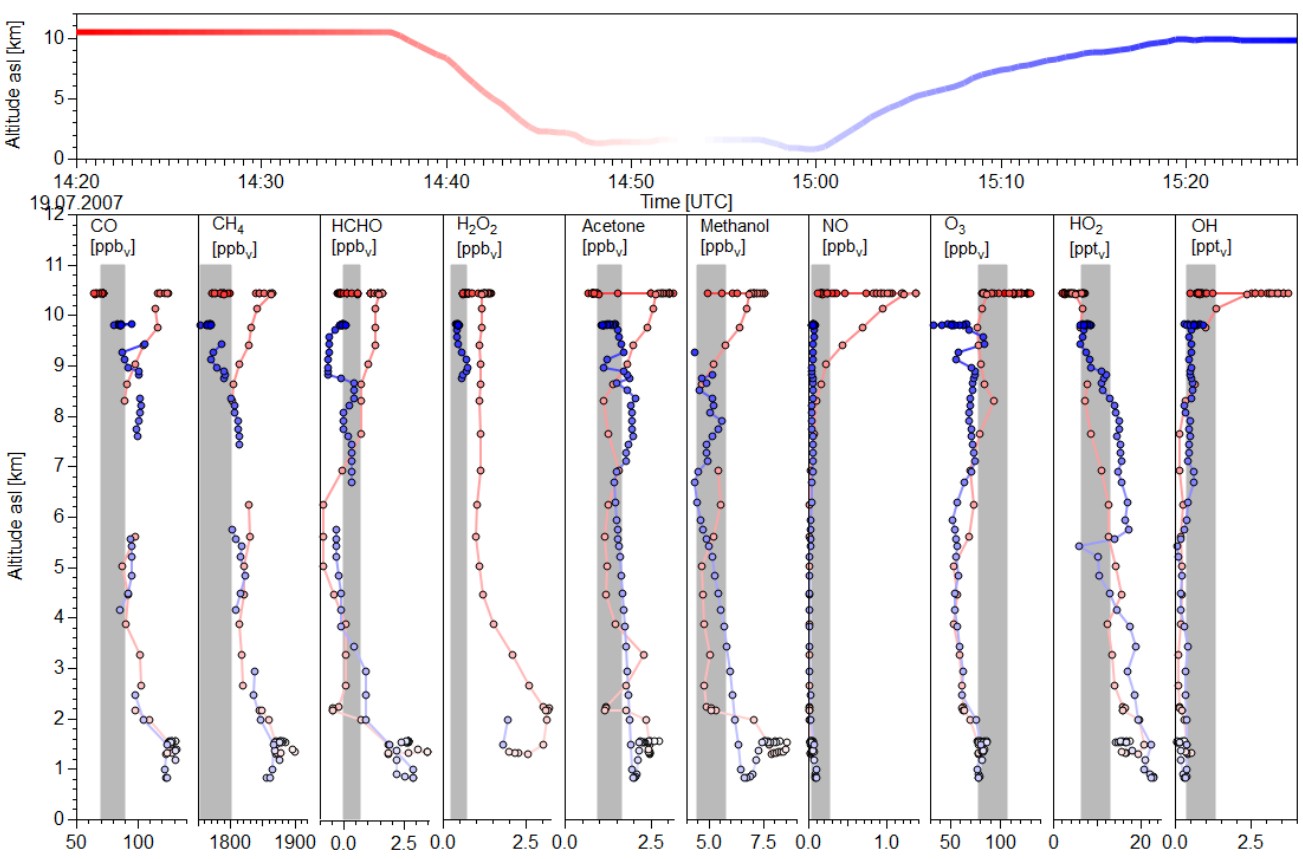


**Figure 7: Vertical profiles from in-situ measurements on 19 July 2007. Colour coding indicates altitude (see upper panel). The grey shaded area shows typical mixing ratios in the upper troposphere.**





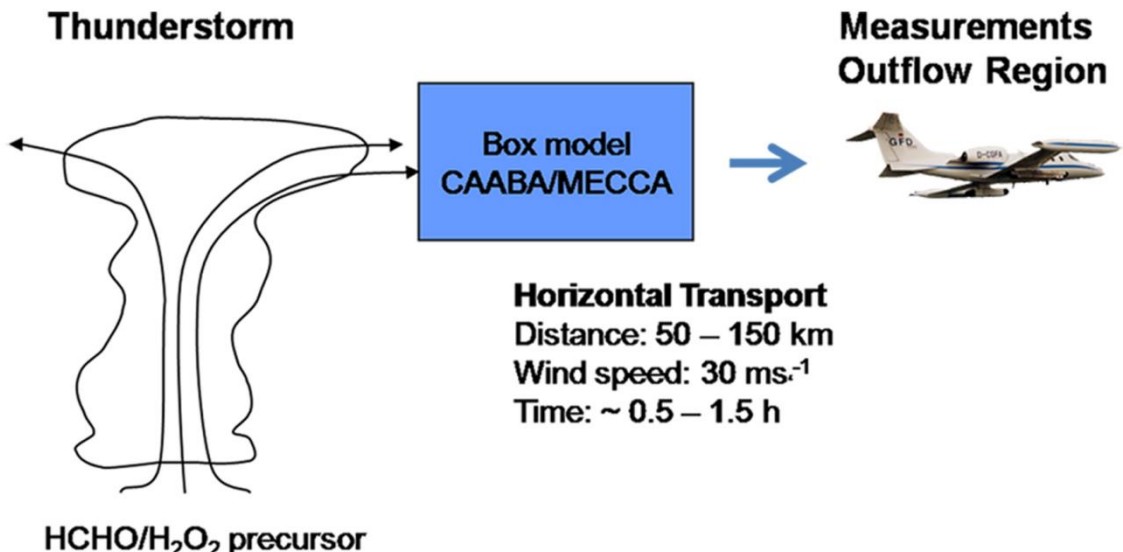


**Figure 8: Scheme for the initialisation of the chemical box model studies.**


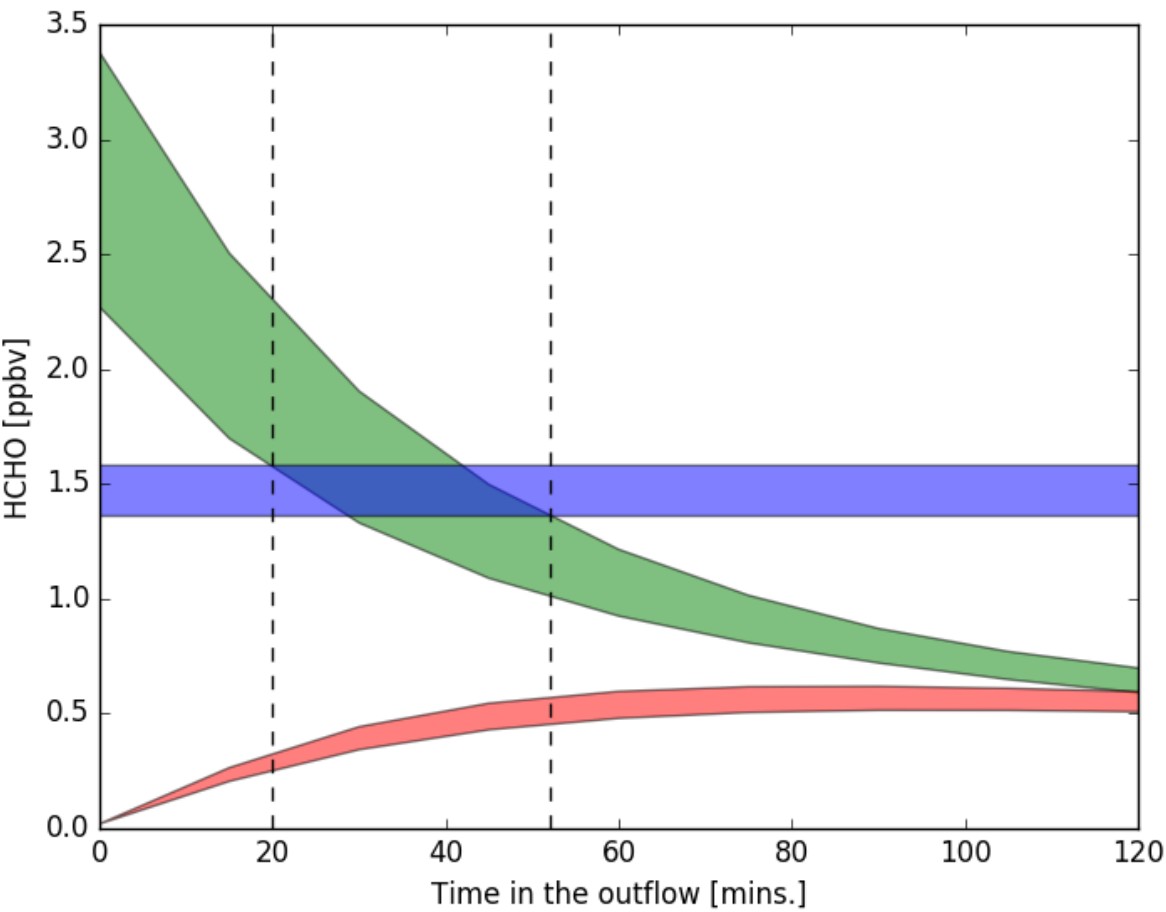


**Figure 9: (a): Simulated temporal evolution of HCHO mixing ratios in the outflow region. Coloured bars show the variance from sensitivity studies with randomly varied rate constants and concentrations. The blue bar indicates the range of observed values in the outflow. The red curve simulates photochemical production assuming zero transport from the inflow area, while the green curve shows photochemical degradation assuming 100 % transport from the inflow area. The dashed vertical lines indicate the**
**best fits for the processing time.**





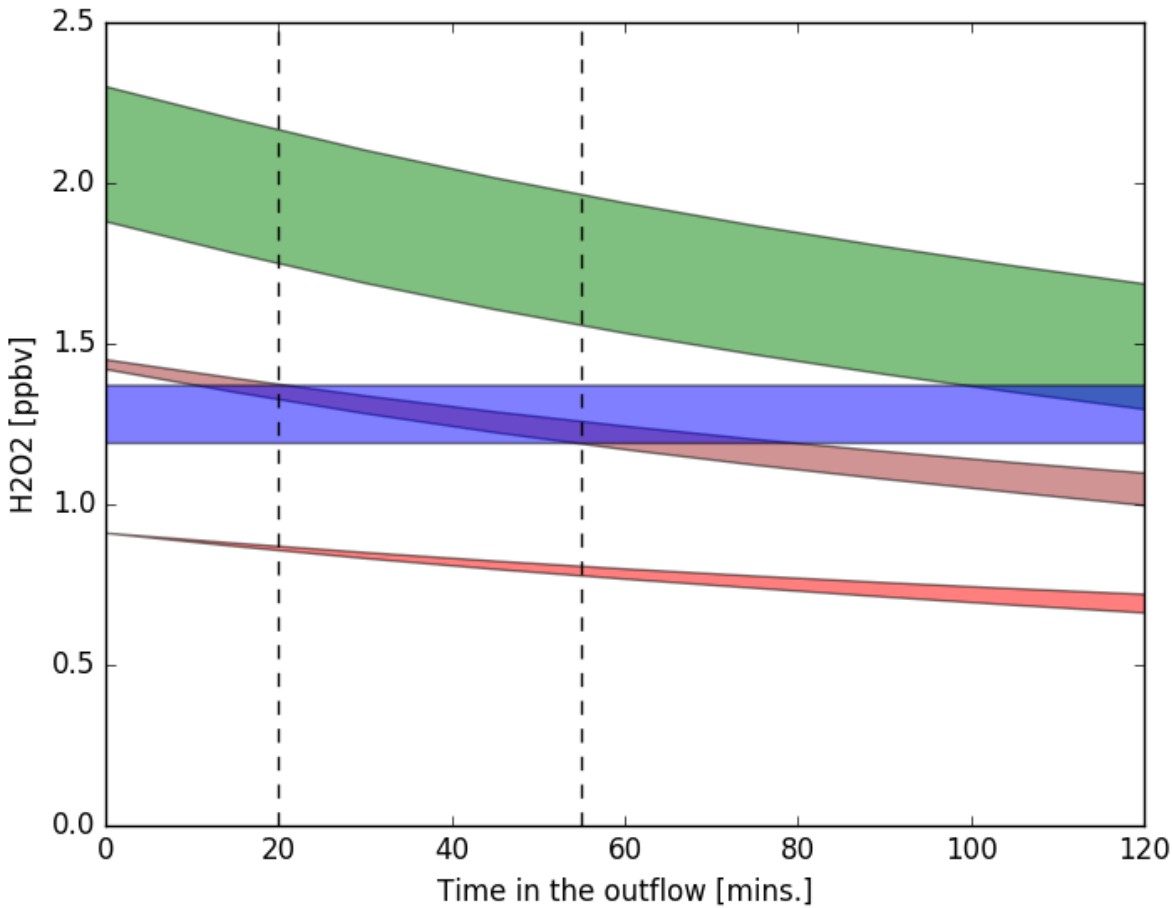

**Figure 10: Simulated temporal evolution of H₂O₂ mixing ratios in the outflow region. Coloured bars show the variance from sensitivity studies with randomly varied rate constants and concentrations. The blue bar indicates the range of observed values in the outflow. The red curve simulates photochemical change assuming zero transport from the inflow area, while the green curve shows photochemical degradation assuming 100 % transport from the inflow area. The brown curve is the best fit for processing times deduced from the HCHO study (dashed vertical lines).**
