# Peer review of "The influence of deep convection on HCHO and H2O2 in the upper troposphere over Europe"

_Atmospheric Chemistry and Physics, 2017_

## Referee Comment (RC2)

Review of Bozem et al., *The influence of deep convection on HCHO and H$_2$O$_2$ in the upper troposphere over Europe.*

**Summary and General Comments:** Bozem and coauthors present aircraft measurements of a wide range of reactive trace gases made in the outflow of deep convection. Their observations focus on the convective redistribution of two soluble HO$_x$ precursors, HCHO and H$_2$O$_2$. Using measurements made in the inflow and outflow region of a single storm, Bozem et al calculate that HCHO and H$_2$O$_2$ are transported with high efficiency by deep convection. The manuscript describes new results and is interpreted in the context of a photochemical model. I have a series of comments that would need to be addressed prior to publication in ACP:

1) Prior measurement campaigns have used the ratio of H$_2$O$_2$:CH$_3$OOH as an indicator of fresh convection due to the preferential scavenging of H$_2$O$_2$. Are measurements of CH$_3$OOH available from this flight to comment on this approach? Based on the scavenging efficiencies reported here, one would expect that ratio not to be very sensitive, yet it has been shown to work well in the remote Pacific.

2) The authors suggest that PBL air is transported to the UT and detrained into the UT undiluted. This seems very hard to believe. Prior aircraft studies have calculated that this ratio is of order 0.2 (from measurements of CO, CH$_4$, CO$_2$, C$_2$H$_6$, and CH$_3$OH (Bertram et al., 2007). Modelling studies have calculated this ratio to be closer to 0.25 (Mullendore et al., 2005). This suggests that convectively lofted PBL air is rapidly mixed on ascent or during detrainment into the UT. The authors should comment in more detail on how their measurements fit in the context of prior measurements since this is an important component of the measured scavenging efficiency for HCHO and H$_2$O$_2$.

**Specific Comments:**

Line 8: " Earth's"

Line 71: "project included  a total of"

Line 72: Give the country (Germany) of Hohn as is done for Corsica and Kiruna

Section 2.2: Was the altitude (temperature and pressure) of convective detrainment used to drive MECCA?

Line 139: What is the evidence for this? This would be an average updraft velocity of about 1 m s$^{-1}$. This is reasonable, but I am curious how/if this was measured.

Line 209: Was the 30 ms$^{-1}$ horizontal wind speed measured? How sensitive are the model conclusions to this number.

Section 3.3: What time of day was the model initiated? At the time of convective detrainment? This, of course, makes a strong difference in photolysis and chemical lifetimes.

Line 310 and beyond: It would be helpful to be consistent in using either scavenging efficiency or retention coefficient.

---

## Referee Comment (RC1) · Anonymous Referee #2 · 12 Apr 2017

This is an interesting study that represents important research that should be of interest to readers of this journal. The results of this research will add to a growing list of studies dealing with the important topic of convective transport of reactive radical precursors to the UT and LS. This paper is within the scope of ACP and meets the scientific quality of this journal. However, after having said this, this reviewer has some major concerns regarding the results as presented and would like to see more supporting evidence in the areas discussed below. Accordingly, this reviewer believes this paper should be accepted for publication after some **major revisions** are made.

First, CO and $CH_4$ may not be the best species to account for entrainment/dilution during both vertical transport as well as horizontal transport out of the anvil. As shown, the contrast in mixing ratios between the convective outflow (OF), the free troposphere, and the boundary layer inflow (IF) are small and thus information on the entrainment rates may not be reliable. Measurement of other species with much more dynamic vertical profiles like various hydrocarbons would be preferable in determining entrainment rates. Can the authors employ their canister measurements of hydrocarbons like i/n butane and i/n pentane and their ratios to address this as well as to further verify that the outflow is coherently related to the inflow?

Nevertheless, given the near unity CO OF/IF ratio of 0.93, one cannot assume that entrainment does not exist. It's hard to imagine there is no entrainment dilution during convective transport from the BL to ~ 10 km, followed by no entrainment dilution of UT background air during the OF. Can these assumptions be wrong? Very similar CO OF/IF ratios were measured during DC3, and yet entrainment was still found to be important. In fact, using your CO IF and OF values in the altitude dependent entrainment model of Fried et al. [2016] with estimates of your background CO values in each 1km altitude bin, I get a net entrainment rate of 3.6%/km. Using this entrainment rate, I calculate that the HCHO value at the storm core should be ~ 2.054 ppbv, which should then be used to compare with your 1.45 ppb OF value, which has to be further modified for production and destruction. Even though my calculations are crude (mixing together entrainment from vertical transport and horizontal outflow in the anvil), they serve to illustrate that dilution of background air should not be ignored.

However, my second and biggest concern relates to the appropriate IF values to use for this analysis. The authors attempt to address this in their discussion section on page 11, by stating that although it is not possible to unambiguously identify the inflow area, their HCHO and H2O2 boundary layer mixing ratios near Dresden are similar to other boundary layer observations during HOOVER II and should thus be representative of the convective IF values. However, as shown by Fried et al. [2016] if this assumption does not hold, then one can obtain both higher (~80%) and lower (~20%) HCHO scavenging efficiencies (SEs), depending upon the circumstances. In particular, large changes in boundary layer isoprene mixing ratios, as one example, can cause erroneous HCHO SE determinations unless one can be certain that the inflow is related to the outflow. The authors need to provide more convincing evidence to this effect in order to reconcile if their much lower SEs with other studies for both HCHO and H2O2 are caused by this or by differences in storm dynamics and microphysics. Do the authors have measurements of isoprene and/or any other sources of HCHO in the boundary layer to help this agrument? Simply invoking differences in ice retention factors cannot explain the lower HCHO and H2O2 SEs in the present

study with the DC3 results. As discussed in the DC3 studies in the case of HCHO, large changes in calculated HCHO ice retention factors from 0.25 (and most recently 0.15) to 1.0 all result in calculated HCHO SEs near 100%. It is only when HCHO is completely degassed from ice (ice retention of 0) can the modeled results reproduce the ~ 50% SE results deduced from measurements.  There is no scenario where changing the ice retention factor produced lower SE results. Likewise, for $H_2O_2$ Bela et al. [2016] and Barth et al. [2016] in their simulations found that with $H_2O_2$ ice retentions $\geqq$ 0.25, the $H_2O_2$ SE approached 100%, and with ice retentions of 0, one obtains $H_2O_2$ SEs of 80% ± 12%. Again, ice retention factors cannot explain the differences.

Therefore, the authors need to seek other explanations for the differences with DC3 results. Can differences in storm dynamics and microphysics be the cause or can differences in IF and OF airmasses be the cause? In the case of the former, the authors should try and contrast differences between the studied storms here and DC3. To eliminate the latter, the authors need to provide more convincing evidence that the IF is related to the OF. In addition, the authors need to raise the possibility that in contrast to most DC3 measurements acquired in the anvil, the measurements here were obtained in clear air and this may allow the hydrometeors a chance to evaporate, thereby degassing the dissolved species resulting in low SEs.

---

## Author Comment (AC1) · 14 Jul 2017

**We thank the referee for her/his helpful comments that we will address in the revised manuscript.**

This is an interesting study that represents important research that should be of interest to readers of this journal. The results of this research will add to a growing list of studies dealing with the important topic of convective transport of reactive radical precursors to the UT and LS. This paper is within the scope of ACP and meets the scientific quality of this journal. However, after having said this, this reviewer has some major concerns regarding the results as presented and would like to see more supporting evidence in the areas discussed below. Accordingly, this reviewer believes this paper should be accepted for publication after some **major revisions** are made.

First, CO and $CH_4$ may not be the best species to account for entrainment/dilution during both vertical transport as well as horizontal transport out of the anvil. As shown, the contrast in mixing ratios between the convective outflow (OF), the free troposphere, and the boundary layer inflow (IF) are small and thus information on the entrainment rates may not be reliable. Measurement of other species with much more dynamic vertical profiles like various hydrocarbons would be preferable in determining entrainment rates. Can the authors employ their canister measurements of hydrocarbons like i/n butane and i/n pentane and their ratios to address this as well as to further verify that the outflow is coherently related to the inflow?

**Answer:**

**We partly agree that CO and in particular $CH_4$ are not ideal tracers to estimate entrainment rates. While the referee's criticism is correct that the dynamical range of $CH_4$ is small throughout the troposphere, this is not necessarily true for CO: median mixing ratios in the inflow and outflow region are 127 ppbv and 118 ppbv, respectively, while the background mixing ratio in the upper troposphere is 63 ppbv and the mixing ratio in the middle troposphere 90 ppbv. Taking into account the variability (instrument's precision and atmospheric variability) these differences are large enough (and significant at least for the contrast between inflow and entrainment regions) to permit a calculation of entrainment rates. Please note that there are also other tracers, e.g. methanol or acetone that can be used for this purpose. NMHC ratios, at least from canister measurements, are not suitable for this purpose in our case. In fact the number of samples is rather limited (24 canisters with a volume of 0.8 l each) and on this particular day two flights were performed, so that only 12 canisters were filled during the second flight on which our study is based. Additionally, one has to take into account that the sampling itself takes more than**

c Author(s) 2017. CC-BY 3.0 License.

[Figure]

**several seconds (lower troposphere) to several tens of seconds in the upper troposphere, so that rather large horizontal and vertical averaging takes place over filling one canister. The data base on NMHC is thus insufficient to perform a similar analysis as done in Fried et al., 2016.**

Nevertheless, given the near unity CO OF/IF ratio of 0.93, one cannot assume that entrainment does not exist. It's hard to imagine there is no entrainment dilution during convective transport from the BL to ~ 10 km, followed by no entrainment dilution of UT background air during the OF. Can these assumptions be wrong? Very similar CO OF/IF ratios were measured during DC3, and yet entrainment was still found to be important. In fact, using your CO IF and OF values in the altitude dependent entrainment model of Fried et al. [2016] with estimates of your background CO values in each 1km altitude bin, I get a net entrainment rate of 3.6%/km. Using this entrainment rate, I calculate that the HCHO value at the storm core should be ~ 2.054 ppbv, which should then be used to compare with your 1.45 ppb OF value, which has to be further modified for production and destruction. Even though my calculations are crude (mixing together entrainment from vertical transport and horizontal outflow in the anvil), they serve to illustrate that dilution of background air should not be ignored.

**Answer:**

**We agree with the referee that the CO (and other species) OF/IF ratio of 0.93 indicates dilution of the outflow to some extent. As pointed out on page 7, line 204, this ratio (and others like methane, acetone and methanol) is not significantly different from unity based on the variability (1-sigma) given in the last column of Table 2. We nevertheless agree that this assumption might be an oversimplification. Therefore we applied a two box model to calculate OF mixing ratios from the Inflow (IN) plus Entrainment (EN) according to**

**OF = x EN + (1-x) IN**

**using values for OF, IN and EN from table 1, 2 and figure 7. We derive the following entrainment rates: 24 % (CO), 26 % (CH$_4$), 30 % (Acetone), 19 % (methanol). Assuming an average value of 25 % entrainment we calculate maximum mixing ratios for HCHO and H$_2$O$_2$ at storm core of 2.05 ppbv and 1.82 ppbv, respectively. Please note that the HCHO value is identical to the estimation made by the referee based on entrainment rates taken from Fried et al., 2016.**

**In the revised manuscript we will add this analysis and use the derived starting values in a sensitivity run of the box model to account for photochemical modification of HCHO and H$_2$O$_2$ in the outflow.**

However, my second and biggest concern relates to the appropriate IF values to use for this analysis. The authors attempt to address this in their discussion section on page 11, by stating that although it is not possible to unambiguously identify the inflow area, their HCHO and H2O2 boundary layer mixing ratios near Dresden are similar to other boundary layer observations during HOOVER II and should thus be representative of the convective IF values. However, as shown by Fried et al. [2016] if this assumption does not hold, then one can obtain both higher (~80%) and lower (~20%) HCHO scavenging efficiencies (SEs), depending upon the circumstances.

**Answer:**

**We agree that establishing a connection between the timing and location of the inflow area and the corresponding outflow of a convective system is the most critical aspect of this study (and**

c Author(s) 2017. CC-BY 3.0 License.

[Figure]

**others, as demonstrated by Fried et al., 2016). Unfortunately, Lagrangian experiments are not possible, so the only way to establish an unambiguous connection between in- and outflow would be through the use of an artificial tracer released in the inflow area, ideally from a second airplane. Here we have to rely on sequential measurements first in the outflow and later in the potential inflow area. Due to the time shift associated with the vertical transport and the movement of the convective system itself it is not possible to determine the correct inflow area. So an inflow area is only representative if it is rather homogeneous with respect to space and time. We interpret the fact that several conservative tracers show similar ratios between in- and outflow as an indication that this assumption is fulfilled here. Additionally, Fig. 6 of Klippel et al. (2011) indicate that HCHO and $H_2O_2$ mixing ratios in the boundary layer are within the range of observations made during all HOOVER II flights in the latitude belt from $50°N – 57.5°N$. Additionally we checked that the wind direction in the boundary layer is such that an inflow into the approaching storm can be assumed, as has been done by Fried et al., 2016. It is not possible to determine the height of the layer from which the inflow takes place. While CO and e.g. some NMHC might be well mixed in the continental boundary layer (CBL), this is not the case for $H_2O_2$ and to some lesser extend HCHO that exhibit strong gradients in the CBL and across the boundary layer (e.g. Klippel et al., 2011). So we agree with the referee that appropriate IF values are the most critical, but we do not see a way to address this question other than we have done.**

In particular, large changes in boundary layer isoprene mixing ratios, as one example, can cause erroneous HCHO SE determinations unless one can be certain that the inflow is related to the outflow.

**Answer:**

**Isoprene concentrations in the in- and outflow area were measured with a PTR-MS (Colomb et al., 2006, doi:10.1071/EN06020). There are only 3 data points yielding 0.13 ppbv (OF) and 0.12 ppbv (IN), which are below the instrument's detection limit. Thus we conclude that isoprene has no significant influence on secondary HCHO formation in the outflow. As shown in our model studies, both HCHO and $H_2O_2$ instead show a tendency for decreasing mixing ratios due to the direct proportionality of the sink term to the concentrations itself.**

The authors need to provide more convincing evidence to this effect in order to reconcile if their much lower SEs with other studies for both HCHO and H2O2 are caused by this or by differences in storm dynamics and microphysics. Do the authors have measurements of isoprene and/or any other sources of HCHO in the boundary layer to help this agrument? Simply invoking differences in ice retention factors cannot explain the lower HCHO and H2O2 SEs in the present study with the DC3 results. As discussed in the DC3 studies in the case of HCHO, large changes in calculated HCHO ice retention factors from 0.25 (and most recently 0.15) to 1.0 all result in calculated HCHO SEs near 100%. It is only when HCHO is completely degassed from ice (ice retention of 0) can the modeled results reproduce the ~ 50% SE results deduced from measurements. There is no scenario where changing the ice retention factor produced lower SE results. Likewise, for H2O2 Bela et al. [2016] and Barth et al. [2016] in their simulations found that with H2O2 ice retentions $\geq$ 0.25, the H2O2 SE approached 100%, and with ice retentions

c Author(s) 2017. CC-BY 3.0 License.

[Figure]

of 0, one obtains H2O2 SEs of 80% ± 12%. Again, ice retention factors cannot explain the differences.

**Answer:**
**Assuming a 50 % scavenging efficiency for HCHO and a mixing ratio of 1.45 ppbv in the OF or 2.05 ppbv estimated at the cloud core would yield an IF mixing ratio of 3 – 4 ppbv. The same calculation for $H_2O_2$ assuming an SE of 80% and a cloud top mixing ratio of 1.25 and 1.82 ppbv, respectively, yields an $H_2O_2$ mixing ratio in the inflow area of 6 - 9 ppbv. Based on the observations of both species during HOOVER II (Klippel et al., 2011) the simultaneous occurrence of these mixing ratios for both species, in particular at the same altitude, is very unlikely. Assuming the scavenging efficiencies determined from the DC3 campaign seem to yield inconsistent results.**

Therefore, the authors need to seek other explanations for the differences with DC3 results. Can differences in storm dynamics and microphysics be the cause or can differences in IF and OF airmasses be the cause? In the case of the former, the authors should try and contrast differences between the studied storms here and DC3. To eliminate the latter, the authors need to provide more convincing evidence that the IF is related to the OF. In addition, the authors need to raise the possibility that in contrast to most DC3 measurements acquired in the anvil, the measurements here were obtained in clear air and this may allow the hydrometeors a chance to evaporate, thereby degassing the dissolved species resulting in low SEs.

**Answer:**
**Differences in the storm dynamics and microphysics between DC3 and HOOVER cannot be investigated since these details are not available for HOOVER. Contrary to DC3 HOOVER was not a coordinated campaign to study convective transport. Here we rely on a single event study, based on the measurements from one aircraft. As pointed out in Fig. 8, and unfortunately not mentioned explicitly in the description of the observations in section 3.2, all measurements were performed in cloud free air. So the referee's suggestion that the differences to DC3 might be due to degassing from evaporating hydrometeors is a potential explanation that we will address in section 4 (Discussion and conclusions) of the revised manuscript.**

---

## Author Comment (AC2) · 14 Jul 2017

**We thank the referee for her/his helpful comments that we will address in the revised manuscript.**

**Summary and General Comments:** Bozem and coauthors present aircraft measurements of a wide range of reactive trace gases made in the outflow of deep convection. Their observations focus on the convective redistribution of two soluble HOx precursors, HCHO and H2O2. Using measurements made in the inflow and outflow region of a single storm, Bozem et al calculate that HCHO and H2O2 are transported with high efficiency by deep convection. The manuscript describes new results and is interpreted in the context of a photochemical model. I have a series of comments that would need to be addressed prior to publication in ACP:

1) Prior measurement campaigns have used the ratio of H2O2:CH3OOH as an indicator of fresh convection due to the preferential scavenging of H2O2. Are measurements of CH3OOH available from this flight to comment on this approach? Based on the scavenging efficiencies reported here, one would expect that ratio not to be very sensitive, yet it has been shown to work well in the remote Pacific.

**Answer:**

**Specified CH$_3$OOH (MHP) measurements were not made. As described in Klippel et al. (2011) the instrument used to measure H$_2$O$_2$ also provides a measurement of organic hydro-peroxides (ROOH). This measurement suffers from the different solubilities of the individual ROOHs (e.g. the solubility of MHP is only 60% of that of H$_2$O$_2$ due to the smaller Henry's law coefficient) and thus changes of the sensitivity for different ROOH. Under the assumption that all ROOH is MHP and correcting for its lower solubility, ROOH measurements can be interpreted as an upper limit for MHP (Klippel et al., 2011). The assumption that all ROOH is MHP is often justified in the free troposphere where MHP is the dominant ROOH, but not necessarily in the boundary layer (Klippel et al., 2011). The average ROOH mixing ratio in the inflow and outflow regions were 0.45 ± 0.02 ppbv and 0.68 ± 0.07 ppbv, respectively. The fact that the MHP concentration is higher in the outflow area than in the inflow indicates that the assumption that all ROOH is MHP is not justified in this case and that the ROOH partitioning most likely changes due to cloud processing. Due to the uncertainty associated with the ROOH measurements we cannot address the question raised by the referee with the available data set.**

c Author(s) 2017. CC-BY 3.0 License.

[Figure]

2) The authors suggest that PBL air is transported to the UT and detrained into the UT undiluted. This seems very hard to believe. Prior aircraft studies have calculated that this ratio is of order 0.2 (from measurements of CO, CH4, CO2, C2H6, and CH3OH (Bertram et al., 2007). Modelling studies have calculated this ratio to be closer to 0.25 (Mullendore et al., 2005). This suggests that convectively lofted PBL air is rapidly mixed on ascent or during detrainment into the UT. The authors should comment in more detail on how their measurements fit in the context of prior measurements since this is an important component of the measured scavenging efficiency for HCHO and H2O2.

**Answer:**

**As pointed out on page 7, line 204 the CO ratio (outflow/inflow) (also methane, acetone and methanol) is not significantly different from unity considering their variability (1-sigma), given in the last column of Table 2, and thus entrainment did not seem to have played a role. In retrospect, this assumption might be an oversimplification. In the revised manuscript we apply a two box model to calculate outflow (OF) mixing ratios from the inflow (IN) and the entrainment (EN) according to**

**OF = x EN + (1-x) IN**

**using values for OF, IN and EN from table 1, 2 and figure 7. We derive the following entrainment rates: 24 % (CO), 26 % (CH₄), 30 % (Acetone), 19 % (methanol), indicating that roughly 75 % of the air in the outflow stems from the boundary layer. Hauf et al., (Hauf, T., P. Schulte, R. Alheit, and H. Schlager (1995), Rapid vertical trace gas transport by an isolated midlatitude thunderstorm, J. Geophys. Res., 100(D11), 22957–22970, doi:10.1029/95JD02324) concluded from a case study of a thunderstorm over Basel (Switzerland) that the cloud contained "protective cores" where air from the boundary layer was transported almost undiluted directly to the anvil. Similar observations were reported by Poulida et al. (Poulida, O., R. R. Dickerson, and A. Heymsfield (1996), Stratosphere-troposphere exchange in a midlatitude mesoscale convective complex: 1. Observations, J. Geophys. Res., 101(D3), 6823–6836, doi:10.1029/95JD03523 ) and Ström et al. (Ström, J., H. Fischer, J. Lelieveld, and F. Schröder (1999), In situ measurements of microphysical properties and trace gases in two cumulonimbus anvils over western Europe, J. Geophys. Res., 104(D10), 12221–12226, doi:10.1029/1999JD900188). These earlier studies corroborate that the ratio of boundary layer air in the outflow can be quite high, as has been found in our study. This question will be addressed in the revised manuscript.**

**Specific Comments:**

Line 8: " Earth's"
**Answer:**
**ok**

Line 71: "project included  a total of"
**Answer:**
**ok**

Line 72: Give the country (Germany) of Hohn as is done for Corsica and Kiruna

c Author(s) 2017. CC-BY 3.0 License.

[Figure]

**Answer:**

**ok**

Section 2.2: Was the altitude (temperature and pressure) of convective detrainment used to drive MECCA?
**Answer:**

**yes**

Line 139: What is the evidence for this? This would be an average updraft velocity of about 1 m s$^{-1}$. This is reasonable, but I am curious how/if this was measured.
**Answer:**
**The statement is an interpretation of the series of cloud maps in Figure 5. The updraft velocity was not measured.**

Line 209: Was the 30 ms$^{-1}$ horizontal wind speed measured? How sensitive are the model conclusions to this number.
**Answer:**
**Yes, these are in-situ measurements on-board the Learjet. The model results indicate that the ultimate mixing ratio in particular for HCHO is very sensitive to the processing time. The time estimated from the cloud distance and measured wind speed is similar to the processing time estimated from the model to reach the observed HCHO mixing ratio.**

Section 3.3: What time of day was the model initiated? At the time of convective detrainment? This, of course, makes a strong difference in photolysis and chemical lifetimes.
**Answer:**
**The model was constrained to measured $J(NO_2)$ photolysis rates. All other photolysis rates were calculated with the TUV model and scaled to the measured $J(NO_2)$ to account for cloud effects.**

Line 310 and beyond: It would be helpful to be consistent in using either scavenging efficiency or retention coefficient.
**Answer:**
**In our discussion scavenging efficiency accounts for all processes that ultimately remove soluble species from the gas phase (rain-out, gravitational removal of ice particle, graupel, hail etc.), while the retention coefficient describes the behavior of a soluble gas during the freezing of a rain drop. A retention coefficient of less than 100% thus indicates that some gas is released from the droplet during the freezing process. In the revised manuscript we will make sure that this difference is clearly described.**